# Feature Selection for Huge Data via Minipatch Learning

## Abstract

Feature selection often leads to increased model interpretability, faster computation, and improved model performance by discarding irrelevant or redundant features. While feature selection is a well-studied problem with many widely-used techniques, there are typically two key challenges: i) many existing approaches can become computationally intractable in huge-data settings on the order of millions of features; and ii) the statistical accuracy of selected features often degrades in high-dimensional, high-noise, and high-correlation settings, thus hindering reliable model interpretation. In this work, we tackle these problems by developing Stable Minipatch Selection (STAMPS) and Adaptive STAMPS (AdaSTAMPS). These are meta-algorithms that build ensembles of selection events of base feature selectors trained on many tiny, random or adaptively-chosen subsets of both the observations and features of the data, which are named minipatches. Our approaches are general and can be employed with a variety of existing feature selection strategies and machine learning techniques in practice. In addition, we empirically demonstrate that our approaches, especially AdaSTAMPS, outperform many competing methods in terms of feature selection accuracy and computational time in a variety of numerical experiments; we also show the efficacy of our method in challenging high-dimensional settings common with biological data. Our methods are implemented in the `Python` package `minipatch-learning`.

## 1 Introduction

Feature selection is critical in statistical machine learning for improving interpretability of machine learning models. More formally, let $(\mathbf{y}, \mathbf{X}) \in \mathbb{R}^N \times \mathbb{R}^{N \times M}$ denote the data with $N$ observations each having $M$ features. Assume there exists a subset $S \subset \{1, \ldots, M\}$ such that the response $y_i$ is independent of the features in the complement $S^c$, conditional on features in $S$ (Lu et al., 2018). Hence the goal of feature selection is to accurately infer the set of correct signal features $S$ from the observed data.

Arguably one of the most challenging scenarios for feature selection is in the high-dimensional case (i.e. $N \ll M$) where there is also typically high correlation. While feature selection has been used in a wide variety of applications, there are typically two key challenges that especially arise in high-dimensional settings: i) many existing techniques can quickly become computationally intractable in huge-data settings on the order of millions of features; and ii) the statistical accuracy of selected features often degrades with huge numbers of features and in the presence of noise and/or highly correlated features, thus hampering reliable model interpretation.

In this work, our primary goal is to develop practical feature selection frameworks that can select statistically accurate features in a computationally efficient manner, even with huge data in high-dimensional, high-noise, high-correlation settings. We place great emphasis on selecting the correct features, which is a statistical accuracy problem, because doing so would considerably enhance explainability of machine learning methods, enable practitioners to reliably interpret models, and provide insights into the underlying data-generating processes (e.g. candidate gene study (Kohannim et al., 2012)). As stressed in Lipton (2018), interpretability of models is critical in many machine learning applications. Furthermore, we seek to design our feature selection framework to be conducive to efficient computation so that it can be applied in huge-data settings with

hundreds of thousands of features and/or observations. Data of such scale is observed in online marketing, genetics, neuroscience, and text mining, among many other areas of research.

## 1.1 Related Literature

Popular feature selection methods can be divided into three categories - filter, wrapper, and embedded methods (Guyon & Elisseeff, 2003). Filter methods select features based solely on observed characteristics of available data independently of a learning algorithm (Jovic et al., 2015). While filter methods are fast computationally and they might work well for prediction by selecting features related to the output, filter methods usually perform suboptimally in terms of statistical accuracy of selecting the correct features that would be critical for reliable model interpretation because they tend to select sets of correlated, redundant features together. Unlike filter methods, wrapper methods assess the quality of selected features using performance of a predefined learning algorithm. Therefore, wrappers are typically much slower computationally than filter methods (Jovic et al., 2015; Pirgazi et al., 2019). Despite generally better performance than filter methods, wrappers tend to perform suboptimally in selecting statistically accurate features because they are inherently greedy methods.

Another line of work focuses on embedded methods, which can be further divided into optimization-based methods and methods that select features based on properties of the learner. The Lasso (Tibshirani, 1996) is arguably one of the most widely-used optimization-based feature selection methods. The Lasso tends to do better on statistical accuracy of selected features and is model selection consistent under theoretical conditions including the Irrepresentable Condition (Zhao & Yu, 2006; Meinshausen & Bühlmann, 2006). Yet, such theoretical conditions requiring low correlation among features will be less likely to ever hold in high-dimensional settings in which features tend to become increasingly correlated. Additionally, choosing the right amount of regularization for accurate feature selection is challenging in practice (Meinshausen & Bühlmann, 2010). Fitting the Lasso with model selection procedures (e.g. cross-validation (Shao, 1993)) can be computationally challenging in huge-data settings.

Last but not least, various tree-based algorithms such as Random Forest (RF) (Breiman, 2001) constitute a great proportion of embedded methods that select features using properties of the learner. In particular, numerous importance scores such as the RF permutation importance are used to rank features based on their contributions to improving predictive performance of the learner. However, previous studies have found many such importance measures can be biased and even become unstable in high-dimensional, high-correlation settings (Strobl et al., 2007; Nicodemus & Shugart, 2007; Genuer et al., 2010), thus hindering reliable model interpretation. Additionally, features selected using properties of the learner are often sensitive to the choice of tuning hyperparameters, and systematic hyperparameter search poses great computational challenges in huge-data settings.

The idea of randomly subsampling observations and/or features for model training has appeared in many parts of machine learning, including various ensemble learning techniques (Breiman, 1996; 2001; Freund & Schapire, 1997; Louppe & Geurts, 2012; Gomes et al., 2019; LeJeune et al., 2020; Yao et al., 2021; Toghani & Allen, 2021) and the dropout technique (Srivastava et al., 2014) combined with stochastic gradient methods (Hardt et al., 2016) in training deep neural networks. Additionally, there is another line of work focusing on using various data randomization techniques to select more "stable" features for high-dimensional linear regression (Bach, 2008; Meinshausen & Bühlmann, 2010; Wang et al., 2011; Shah & Samworth, 2013; Yu, 2013). More recently, Beinrucker et al. (2016) and Staerk et al. (2021) propose interesting extensions of such stability selection approaches. We seek to leverage these ideas and push them to their full generality by developing practical feature selection frameworks that exploit tiny, (adaptively-chosen) random subsets of both the observations and features of the data for considerably enhancing feature selection accuracy and computational efficiency, hence improving model interpretability.

**Contributions** We summarize our contributions as follows: We develop practical feature selection methods named STAMPS and AdaSTAMPS that leverage random data perturbation as well as adaptive feature sampling schemes to select statistically accurate features with considerably improved computational efficiency (see Section 2), even in high-dimensional settings. Our proposed approaches are general meta-algorithms that can be employed with a variety of existing feature selection strategies and machine learning techniques in practice. In addition, we empirically demonstrate the practical effectiveness of our approaches on multiple synthetic data

sets and a real-world biological data set in Section 3, showing that our frameworks, especially AdaSTAMPS, outperform many competing methods in terms of both feature selection accuracy and computational time in a variety of scenarios. Last but not least, we provide implementations of our methods in the open-source `Python` package `minipatch-learning` (see Section 2.5), available at https://github.com/DataSlingers/minipatch-learning.

## 2 Minipatch Feature Selection

### 2.1 Review: Minipatch Learning

People have used random subsets of observations and features for model fitting in many parts of machine learning (see Section 1). Notably in the ensemble learning literature, some have called tiny, random subsets of both observations and features "minipatches" (Yao et al., 2021; Toghani & Allen, 2021). Using tiny minipatches can yield major computational advantages for huge data. Following the definitions from Yao et al. (2021), we give a brief review of the idea of minipatch: given a pair of response vector $\mathbf{y} \in \mathbb{R}^N$ and data matrix $\mathbf{X} \in \mathbb{R}^{N \times M}$ that consists of $N$ observations each having $M$ features, a minipatch can be obtained by simultaneously subsampling $n$ rows (observations) and $m$ columns (features) without replacement using some form of randomization, typically with $n \ll N$ and $m \ll M$.

### 2.2 Stable Minipatch Selection (STAMPS)

Inspired by stability selection (Meinshausen & Bühlmann, 2010; Shah & Samworth, 2013; Beinrucker et al., 2016; Staerk et al., 2021) that keeps "stable" features based on selection frequencies as well as the idea of minipatch learning (Yao et al., 2021; Toghani & Allen, 2021), we propose a general and practical approach to feature selection named Stable Minipatch Selection (STAMPS). STAMPS is a flexible meta-algorithm that can be employed with a variety of existing feature selection strategies and machine learning techniques.

Our proposed STAMPS method is summarized in Algorithm 1. Here, $\mathbf{y}_{I_k}$ denotes the subvector of $\mathbf{y}$ containing its elements indexed by $I_k$. Similarly, $\mathbf{X}_{I_k, F_k}$ denotes the submatrix of $\mathbf{X}$ containing its rows indexed by $I_k$ and its columns indexed by $F_k$. For brevity, $[M]$ denotes the set $\{1, 2, \ldots, M\}$.

Specifically, STAMPS fits arbitrary base feature selectors to many tiny, random minipatches and calculates feature selection frequencies by taking an ensemble of estimated feature supports over these minipatches. We define the selection frequency of the $j^{\text{th}}$ feature at the $k^{\text{th}}$ iteration, $\hat{\Pi}_j^{(k)}$, to be the number of times it is sampled and then selected by base feature selectors divided by the number of times it is sampled into minipatches after $k$ iterations. STAMPS eventually outputs a set of stable features $\hat{S}^{\text{stable}}$ whose final selection frequencies are above a user-specific threshold $\pi_{\text{thr}} \in (0, 1)$. We discuss the choice of $\pi_{\text{thr}}$ in Section 2.4.

---

**Algorithm 1** STAMPS

**Input:** $\mathbf{y} \in \mathbb{R}^N$, $\mathbf{X} \in \mathbb{R}^{N \times M}$, $n$, $m$, $\pi_{\text{thr}} \in (0, 1)$
**Initialize:** $\hat{\Pi}_j^{(0)} = 0$, $\forall j \in [M]$
**while** stopping criterion not met **do**

    1) Sample a minipatch: subsample $n$ observations $I_k \subset [N]$ and $m$ features $F_k \subset [M]$ uniformly at random without replacement to get a minipatch $(\mathbf{y}_{I_k}, \mathbf{X}_{I_k, F_k}) \in \mathbb{R}^n \times \mathbb{R}^{n \times m}$

    2) Fit a base selector to minipatch $(\mathbf{y}_{I_k}, \mathbf{X}_{I_k, F_k})$ to obtain an estimated feature support $\hat{\mathcal{S}}_k \subseteq F_k$

    3) Ensemble feature supports $\{\hat{\mathcal{S}}_t\}_{t=1}^k$ to update selection frequencies $\hat{\Pi}_j^{(k)}, \forall j \in [M]$:

$$\hat{\Pi}_j^{(k)} = \frac{\sum_{t=1}^k \mathbb{1}(j \in F_t, j \in \hat{\mathcal{S}}_t)}{\max(1, \sum_{t=1}^k \mathbb{1}(j \in F_t))}$$

**end while**
**Output:** $\hat{S}^{\text{stable}} = \left\{ j \in [M] : \hat{\Pi}_j^{(K)} \geq \pi_{\text{thr}} \right\}$

---

To avoid a forever STAMPS situation, one should employ some type of stopping criterion for Algorithm 1. While there are many possible choices, we happen to use a simple one that is effective in our empirical studies - stop the algorithm if the rank ordering of the top $\min(\max(|\mathcal{H}|, \tau_\mathrm{l}), \tau_\mathrm{u})$ features in terms of selection frequencies $\{\hat{\Pi}_j^{(k)}\}_{j=1}^M$ remain unchanged for the past 100 iterations, where $\mathcal{H} = \{j \in [M] : \hat{\Pi}_j^{(k)} \geq 0.5\}$. Here, $\tau_\mathrm{l}$ and $\tau_\mathrm{u}$ are fixed, user-specific parameters. We suggest setting both to well exceed the approximate number of true signal features, with $\tau_\mathrm{u}$ being an integer multiple of $\tau_\mathrm{l}$.

### 2.3 Adaptive Stable Minipatch Selection (AdaSTAMPS)

We propose to adaptively sample minipatches in ways that further boost computational efficiency as well as feature selection accuracy. Replacing uniform feature sampling in step 1 of Algorithm 1 with any appropriate adaptive feature sampling scheme, we obtain our Adaptive STAMPS (AdaSTAMPS) framework for feature selection.

While there are many possible adaptive sampling techniques to choose from, we develop two types of example algorithms in this work. We describe an exploitation and exploration scheme inspired by the multi-armed bandit literature (Bouneffouf & Rish, 2019; Slivkins, 2019) here in Algorithm 2. For brevity, we use the shorthand AdaSTAMPS (EE) to denote the specific instance of AdaSTAMPS that employs Algorithm 2 as its adaptive feature sampling scheme. In addition, we give another probabilistic adaptive feature sampling scheme in Algorithm 3 in the Appendix, which we abbreviate as AdaSTAMPS (Prob). We use these two in our empirical studies just to show examples of what can be done with our flexible AdaSTAMPS approach. However, one can use many other more sophisticated sampling techniques in practice. We save the investigation of various other adaptive sampling schemes for future work.

In Algorithm 2, $k$ is the iteration counter, $E$ denotes the total number of burn-in epochs, $\{\gamma^{(k)}\}$ is an increasing geometric sequence in the range $[0.5, 1]$ for controlling the trade-off between exploitation and exploration of the input feature space, and $\pi_\mathrm{active} \in (0, 1)$ is a fixed, user-specific threshold for determining the active feature set. We found setting $\pi_\mathrm{active} = 0.1$ works well for many problems in practice.

---

**Algorithm 2** Adaptive Feature Sampling Scheme: Exploitation and Exploration (EE)

---

**Input:** $k$, $M$, $m$, $E$, $\{\hat{\Pi}_j^{(k-1)}\}_{j=1}^M$, $\{\gamma^{(k)}\} = \mathrm{geom\_seq}([0.5, 1])$, $\pi_\mathrm{active}$

**Initialize:** $G = \lceil \frac{M}{m} \rceil$, $\mathcal{J} = \{1, \ldots, M\}$

**if** $k \leq E \cdot G$ **then** // Burn-in stage

  **if** $\mathrm{mod}_G(k) = 1$ **then** // A new epoch

    1) Randomly reshuffle feature index set $\mathcal{J}$ and partition into disjoint sets $\{\mathcal{J}_g\}_{g=0}^{G-1}$
  **end if**

  1) Set $F_k = \mathcal{J}_{\mathrm{mod}_G(k)}$

**else** // Adaptive stage

  1) Update active feature set:

$$\mathcal{A} = \left\{ j \in \{1, \ldots, M\} : \hat{\Pi}_j^{(k-1)} \geq \pi_\mathrm{active} \right\}$$

  2) Exploitation step: Sample $\min(m, \gamma^{(k)}|\mathcal{A}|)$ features $F_{k,1} \subseteq \mathcal{A}$ uniformly at random

  3) Exploration step: Sample $(m - \min(m, \gamma^{(k)}|\mathcal{A}|))$ features $F_{k,2} \subseteq \{1, \ldots, M\} \setminus \mathcal{A}$ uniformly at random

  4) Set $F_k = F_{k,1} \cup F_{k,2}$
**end if**
**Output:** $F_k$

---

During the initial burn-in stage of Algorithm 2, AdaSTAMPS (EE) explores the entire input feature space by ensuring each feature is sampled into minipatches for exactly $E$ times, once for each epoch. The selection frequency of each feature is continuously updated as feature selection is repeatedly conducted on each random

minipatch. After $E$ epochs, Algorithm 2 transitions to the adaptive stage during which it exploits a subset of features that are likely to be true signal features while adaptively exploring the rest of the feature space. Specifically, features with selection frequencies above $\pi_{\text{active}}$ are kept in the active feature set $\mathcal{A}$, which is likely a superset of most of the true signal features. As $k$ increases and feature selection frequencies keep polarizing, $\mathcal{A}$ continues to shrink and hone in on the set of true signal features. To exploit and validate the relatively stable features in $\mathcal{A}$, Algorithm 2 samples an increasing proportion of $\mathcal{A}$ into each new minipatch by gradually increasing $\gamma^{(k)}$. Meanwhile, Algorithm 2 continues to explore the rest of the feature space by drawing from $\{1, \ldots, M\} \setminus \mathcal{A}$ into the minipatch, because a fraction of true signal features might still be outside of $\mathcal{A}$ due to noise and correlation among features. Intuitively, the ever-changing active feature set $\mathcal{A}$ and $\{\gamma^{(k)}\}$ work together to adaptively strike a balance between exploitation and exploration of the input feature space, thus further enhancing computational efficiency and feature selection accuracy (e.g. Figure 4 in the Appendix).

### 2.4 Practical Considerations

Our STAMPS and AdaSTAMPS are general meta-algorithms and they have three tuning hyperparameters: minipatch size ($n$ and $m$) and threshold $\pi_{\text{thr}}$. In general, our methods are fairly robust to the choice of these hyperparameter values. In our empirical studies in Section 3, we use a default threshold $\pi_{\text{thr}} = 0.5$, which generally works well in practice. We also propose a data-driven way to choose $\pi_{\text{thr}}$ as detailed in Algorithm 4 in the Appendix. For choosing the minipatch size, we have empirical studies in Section B.3 of the Appendix investigating how feature selection accuracy and runtime vary for various $n$ and $m$ values. In general, we found taking $m$ to well exceed the number of true signal features (e.g. 3-10 times the number of true signal features) and then picking $n$ relative to $m$ so that it well exceeds the sample complexity of the base feature selector used in step 2 of Algorithm 1 works well in practice. Our empirical results also reveal that our meta-algorithms have largely stable performance for a sensible range of $n$ and $m$ values.

### 2.5 Software Package

To facilitate the application of our proposed methods to more data problems in practice, we also develop an open-source `Python` software package providing high-quality implementations of these meta-algorithms with emphasis on performance and ease of use. This section gives an overview of this software package named `minipatch-learning`, available at https://github.com/DataSlingers/minipatch-learning.

As our proposed methods are designed to tackle large-scale data problems, care has been taken to optimize computational performance of the corresponding implementations in `minipatch-learning`. For instance, careful integration of our adaptive learning schemes with efficient sampling functionality from `numpy` (Harris et al., 2020) ensures a high degree of computational efficiency during code execution. In addition, we strive to optimize the software implementations for memory efficiency by leveraging various efficient data structures such as `numpy`'s view-based model and minimizing unnecessary memory copies. Out-of-scope memory objects are also timely removed to free up more system memory during code execution. Furthermore, we design `minipatch-learning` to be user-friendly. First and foremost, we implement `scikit-learn`-compatible Application Programming Interface (API) for the class objects contained in `minipatch-learning`, making it easier for users to start applying our software, especially if they are already familiar with `scikit-learn` (Pedregosa et al., 2011). Additionally, we develop detailed API documentations for all the class objects and class methods included in `minipatch-learning`. Following the established `numpy` documentation standards, these API documentations provide both precise specifications (e.g. data types) and intuitive explanations for the parameters and software-specific arguments. Last but not least, concise code examples are provided to illustrate the usage of the software package.

## 3 Empirical Studies

In this section, we evaluate our proposed STAMPS and AdaSTAMPS approaches on both synthetic and real-world data sets. Because many people study the problem of feature selection in the linear regression setting and it is also easier to simulate synthetic benchmark data from linear models for comparisons with

existing methods, we first focus on feature selection in the high-dimensional linear regression setting in Section 3.1. However, our proposed feature selection frameworks are not limited to either linear or regression settings. We showcase the efficacy of our AdaSTAMPS method on a real-world classification problem in Section 3.2.

### 3.1 Feature Selection in High-Dimensional Linear Regression

We compare the feature selection accuracy as well as computational time of our proposed methods with a variety of widely-used existing approaches including the Lasso (Tibshirani, 1996), Complementary Pairs Stability Selection (CPSS) (Shah & Samworth, 2013), Randomized Lasso (Meinshausen & Bühlmann, 2010), univariate linear regression filter with False Discovery Rate (FDR) control via the Benjamini-Hochberg procedure (Benjamini & Hochberg, 1995), recursive feature elimination (RFE) (Guyon et al., 2002), and thresholded Ordinary Least Squares (OLS) (Giurcanu, 2016). Our focus is on high-dimensional cases with $N \ll M$ because these settings are arguably the most difficult for feature selection. In this section, we consider three high-dimensional scenarios with the following data matrices $\mathbf{X} \in \mathbb{R}^{N \times M}$:

- **Scenario 1 (S1)** Autoregressive Toeplitz design: the $M = 10000$-dimensional feature vector follows a $\mathcal{N}(\mathbf{0}, \boldsymbol{\Sigma})$ distribution, where $\Sigma_{ij} = \rho^{|i-j|}$ with $\rho = 0.95$; sample size $N = 5000$.

- **Scenario 2 (S2)** Subset of The Cancer Genome Atlas (TCGA) data: this data set ($N = 2834$, $M = 10000$) contains 450K methylation status of the top 10000 most variable CpGs for 2834 patients. This scenario is even more challenging than S1 due to strong and complex correlation structure among CpGs. This data set was obtained via `TCGA2STAT` (Wan et al., 2015).

- **Scenario 3 (S3)** Fullsize TCGA data: this data set ($N = 2834$, $M = 335897$) contains 450K methylation status of 335897 CpGs with complete records for 2834 patients.

Following standard simulation procedures (Meinshausen & Bühlmann, 2010), we create the true feature support set $S$ by randomly sampling $S \subset \{1, \ldots, M\}$ with cardinality $|S| = 20$. The $M$-dimensional sparse coefficient vector $\boldsymbol{\beta}$ is generated by setting $\beta_j = 0$ for all $j \notin S$ and $\beta_j$ is chosen independently and uniformly in $[-3/b, -2/b] \cup [2/b, 3/b]$ for all $j \in S$. Finally, the response vector $\mathbf{y} \in \mathbb{R}^N$ can be generated by $\mathbf{y} = \mathbf{X}\boldsymbol{\beta} + \boldsymbol{\epsilon}$ where the noise vector $(\epsilon_1, \ldots, \epsilon_N)$ is IID $\mathcal{N}(0, 1)$. For S1 and S2, we vary the positive real number $b$ to attain various signal-to-noise ratio (SNR) levels. For S3, SNR is fixed at 5.

For each method, commonly-used data-driven model selection procedures are applied to choose tuning hyperparameters and produce a data-driven estimate of the true feature support. In addition, oracle model selection that assumes knowledge of cardinality of the true support $|S|$ is employed with each method, resulting in an oracle estimate of the true feature support. Detailed descriptions of these data-driven and oracle model selection procedures are given in Section C.1 of the Appendix. We evaluate the feature selection accuracy of each method using the F1 Score, which takes values between 0 and 1 with 1 indicating perfect match between the final estimated feature support and the true feature support $S$. For every method, we compute F1 Scores for both data-driven and oracle estimates.

We use `scikit-learn` (Pedregosa et al., 2011) for all Lasso-based methods, univariate filter, and RFE with Ridge estimator. Due to lack of existing software, we faithfully implemented the thresholded OLS as in Giurcanu (2016). For simplicity, our methods (i.e. STAMPS, AdaSTAMPS (EE), and AdaSTAMPS (Prob)) are employed with the same thresholded OLS as the base selector in step 2 of Algorithm 1 using default settings as detailed in the Appendix. All comparisons were run on a VM with 10 Intel (Cascade Lake) vCPUs and 220 GB of memory.

The feature selection accuracy and computational time of various methods from scenarios S1 and S2 are shown in Figure 1. We see that our AdaSTAMPS consistently outperforms most competing methods in terms of feature selection accuracy in both scenarios. While some competitors such as RFE and the Lasso achieve comparable accuracy to AdaSTAMPS with oracle model selection at some SNR levels (Figure 1B, 1E), their performance degrades drastically when their tuning hyperparameters are chosen via widely-used data-driven ways (Figure 1A, 1D). On the other hand, performance of AdaSTAMPS does not deteriorate as much across

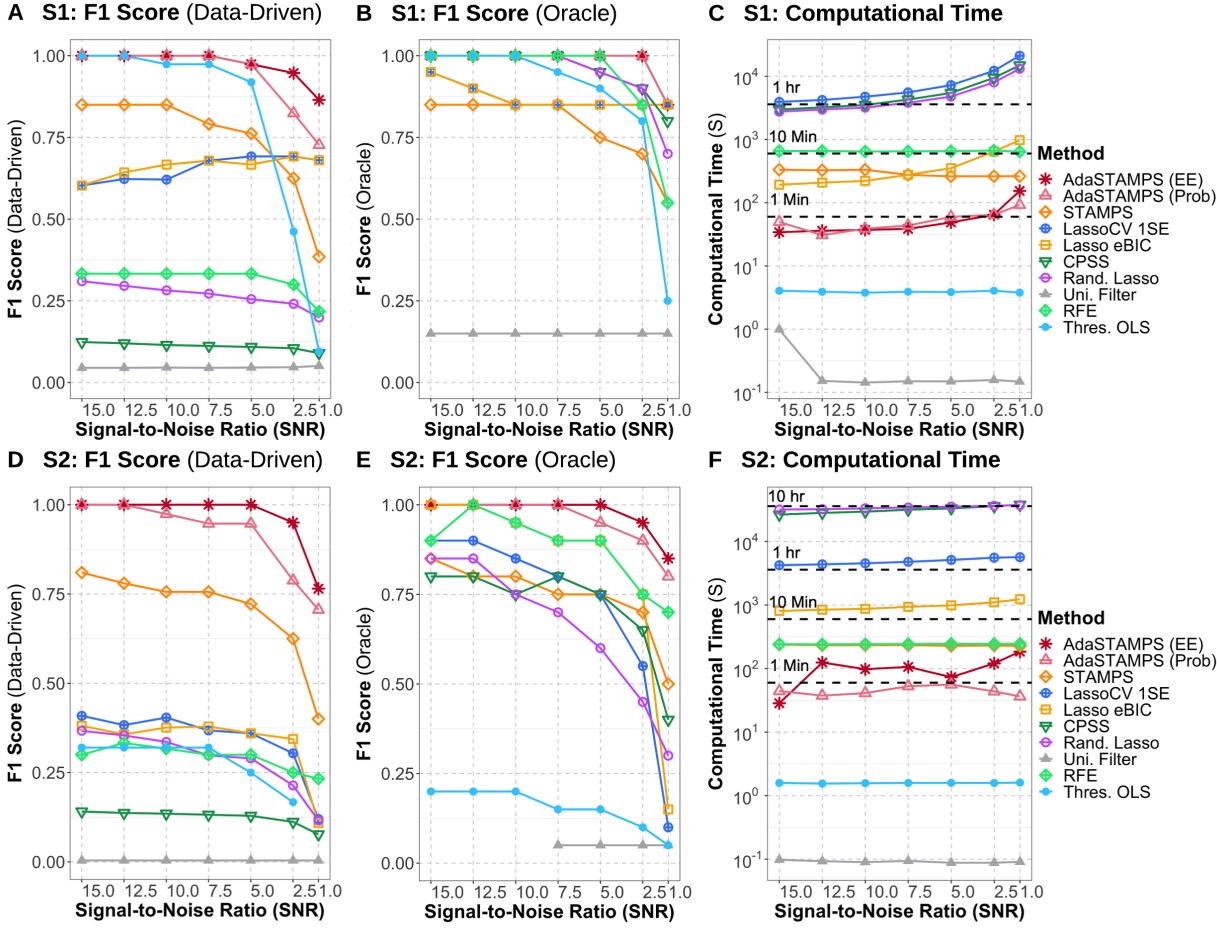

Figure 1: Feature selection accuracy (F1 Score) and computational time from simulation Scenario 1 (S1) and Scenario 2 (S2). (A, D) F1 Score for data-driven feature support estimates; (B, E) F1 Score for oracle feature support estimates; (C, F) Computational time in seconds (logarithmic scale). The fastest runtime is reported for every method with parallelism enabled for methods (LassoCV 1SE, Lasso eBIC, CPSS, Rand. Lasso, RFE) wherever possible. Our AdaSTAMPS outperforms most competitors in terms of feature selection accuracy in both scenarios. Moreover, the runtime of our AdaSTAMPS is among the best of any method with comparable feature selection accuracy.

data-driven and oracle model selection. We also point out that our AdaSTAMPS can potentially boost the accuracy of the existing feature selection strategy used as its base selector - AdaSTAMPS with thresholded OLS as base selector outperforms thresholded OLS itself by a large margin, especially in low SNR settings and in the more challenging scenario S2. Furthermore, AdaSTAMPS is computationally among the fastest of any method with comparable accuracy, with runtime on the order of seconds to minutes as opposed to hours for many competing methods.

The results for the even more challenging scenario S3 are summarized in Table 1. High-dimensional data of such scale is regularly observed in areas such as genetics and neuroimaging. We see that our AdaSTAMPS (EE) perfectly recovers the true feature support across both data-driven and oracle model selection. In addition, STAMPS and AdaSTAMPS (Prob) also perform favorably. On the other hand, most competing methods attain rather low feature selection accuracy even with oracle selection. Clearly, AdaSTAMPS (EE) is more capable of accurately identifying the true signal features even in high-correlation, high-dimensional

Table 1: Results of various methods for simulation Scenario 3 (S3). For each method, the number of selected features determined by data-driven procedures, the data-driven F1 Score, the oracle F1 Score, and computational time in minutes are reported. The method with the best accuracy is bold-faced. Note that the fastest runtime is reported for every method with parallelism enabled for methods (∗) wherever possible. Our AdaSTAMPS outperform all competing methods in terms of feature selection accuracy.

| Method | # Selected Features | F1 (Data-Driven) | F1 (Oracle) | Time (Minute) |
|---|---|---|---|---|
| AdaSTAMPS (EE) | 20 | **1.000** | **1.000** | 19.87 |
| AdaSTAMPS (Prob) | 47 | 0.597 | 0.750 | 52.10 |
| STAMPS | 237 | 0.117 | 0.750 | 74.70 |
| LassoCV 1SE | 194 | 0.112 | 0.600 | 176.60∗ |
| Lasso eBIC | 46 | 0.091 | 0.100 | 61.59∗ |
| CPSS | 81 | 0.238 | 0.500 | 716.94∗ |
| Rand. Lasso | 9 | 0.276 | 0.450 | 838.53∗ |
| Uni. Filter | 277261 | 0.000 | 0.050 | 0.05 |
| RFE | 37035 | 0.001 | 0.350 | 97.82∗ |
| Thres. OLS | 4 | 0.083 | 0.100 | 1.01 |

settings, thus leading to more reliable model interpretation. In the meantime, AdaSTAMPS (EE) takes less than 20 minutes to run, rather than the 14 hours required to solve the Randomized Lasso that achieves the next best data-driven F1 Score among competing methods.

Overall, we empirically demonstrate the effectiveness of our proposed approaches in selecting statistically accurate features with much improved computational efficiency, even in challenging high-dimensional, high-noise, high-correlation scenarios. Even though STAMPS does not perform as well as AdaSTAMPS, STAMPS still performs better than many of the competing methods, especially in terms of data-driven F1 Score (e.g. Figure 1A, 1D). In the meantime, our AdaSTAMPS (both (EE) and (Prob)) largely outperform competing methods in terms of feature selection accuracy while being computationally one of the fastest of any method with comparable accuracy. This suggests that our flexible AdaSTAMPS approach could be a double-win both in terms of statistical accuracy and computation in practice. Intuitively speaking, the strong empirical performance of our proposed methods can be attributed to the following reasons: i) the use of tiny minpatches yields major computational advantages and our adaptive sampling schemes further boost efficiency; ii) the repeated subsampling of features breaks up strong correlations among features that could otherwise derail many existing feature selection strategies; and iii) the final aggregation of selection events over all minipatches reduces variance in a similar manner to that of Random Forest and hence ensures strong performance even in high-noise-low-signal settings.

## 3.2 Real-World Classification Data

We now study the performance of our method when employed as a dimensionality-reduction preprocessing step for downstream classification tasks on a real-world high-dimensional single-cell RNA sequencing (RNASeq) data set from a study of human Glioblastoma (Darmanis et al., 2017). The data set contains expressions of $M = 28805$ genes (features) measured over $N = 3576$ cells (observations). Each cell has a class label corresponding to one of the seven cell types, (*neoplastic*, *oligodendrocyte*, *OPC*, *immune*, *neuron*, *vascular*, *astrocyte*). The data set is split into 70% training and 30% test data via stratified sampling.

Because the identities of true signal features are unknown in such real-world data set, we follow widely-used experimental procedure in the literature (e.g. Chen et al. (2017)) to evaluate feature selection performance. Specifically, we take the top $J$ features selected by a feature selection method on the training data and train a downstream classifier on just these $J$ features to evaluate which method selects more predictive top features. Figure 2 shows the test accuracy of the downstream classifiers versus number of selected features that are used in training these downstream classifiers. Here, we compare AdaSTAMPS with a variety of tree-based wrapper/embedded methods as well as linear model-based methods in Figure 2A and Figure 2B, respectively. For fair comparisons, we use Random Forest (RF) classifier and multinomial regressor with default parameters from `scikit-learn` as the downstream classifiers for the experiments in Figure 2A and

Figure 2B, respectively. Detailed descriptions of the training procedures can be found in Section C.2 of the Appendix.

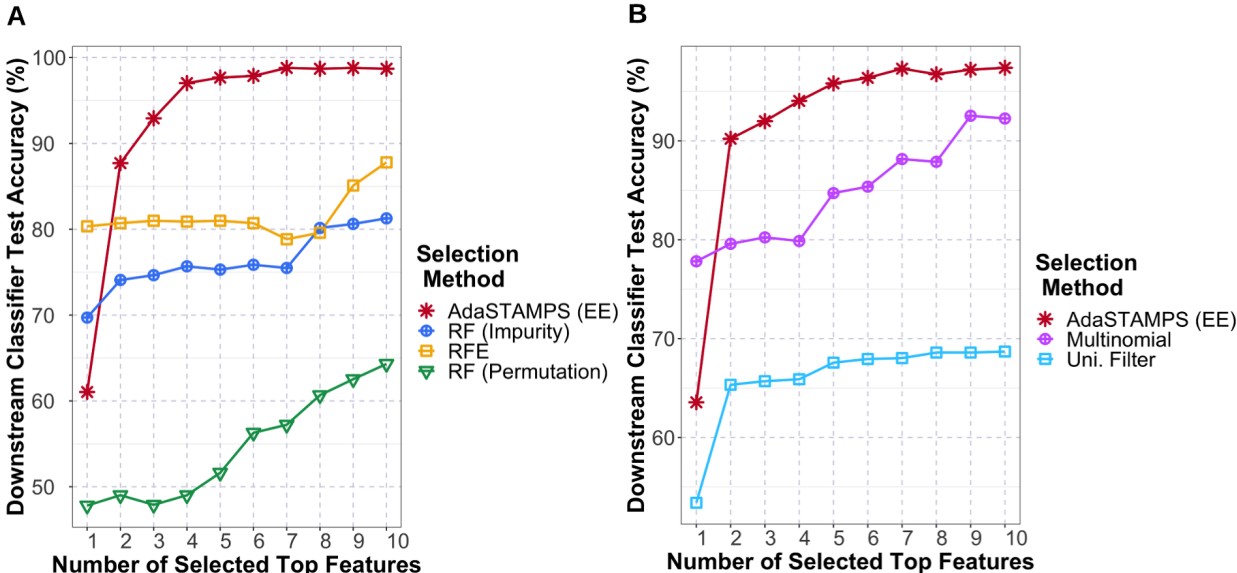

Figure 2: Results for the real-world classification data. We take the top $J$ features selected by a feature selection method and train a downstream classifier on just these $J$ features to evaluate which method selects more predictive top features. (A, B) Test accuracy of these downstream classifiers versus number of selected top features that are used in training the downstream classifiers. RF classifier and multinomial regression are used as the downstream classifiers for (A) and (B), respectively. Overall, the top ten features selected by our AdaSTAMPS (EE) are more informative, leading to better classification accuracy on the test set.

From Figure 2, we see that the downstream classifiers trained on the top ten features selected by AdaSTAMPS (EE) largely achieve better test accuracy than the competitors in both experiments, indicating that the top features identified by AdaSTAMPS (EE) are generally more predictive.

In addition to prediction, unsupervised projection techniques such as the Uniform Manifold Approximation and Projection (UMAP) approach (McInnes et al., 2020) are widely employed in bioinformatics for uncovering useful patterns hidden in data. Here we further demonstrate the utility of the top features selected by AdaSTAMPS for these types of analyses in Figure 3. Specifically, UMAP is fit to the top seven features selected by the AdaSTAMPS selector (above default threshold $\pi_{\mathrm{thr}} = 0.5$) from Figure 2A and the resulting 2-dimensional embeddings are shown in Figure 3B. As a baseline, another UMAP is fit to all $M = 28805$ features and the resulting 2-dimensional embeddings are visualized in Figure 3A. We see that the lower-dimensional embeddings obtained from UMAP fit to the top features selected by AdaSTAMPS give much clearer group patterns and better distinctions between the different cell types. This thus provides further evidence that the top features selected by our AdaSTAMPS method are highly informative and can be useful for a variety of downstream analyses.

Last but not least, we conduct a literature search on the top ten features identified by AdaSTAMPS - we found evidence in the cancer genomics literature, which links at least six of them (i.e. *PTPRZ1*, *CLU*, *APOD*, *LAPTM5*, *IGFBP7*, *EGFR*) to various aspects of glioma biology (Hunter et al., 2005; Pen et al., 2008; Autelitano et al., 2014; Darmanis et al., 2017; Berberich et al., 2020). While this is only a preliminary investigation of the utility of the top features selected by AdaSTAMPS, successfully identifying genes that have been validated in the scientific literature is encouraging evidence.

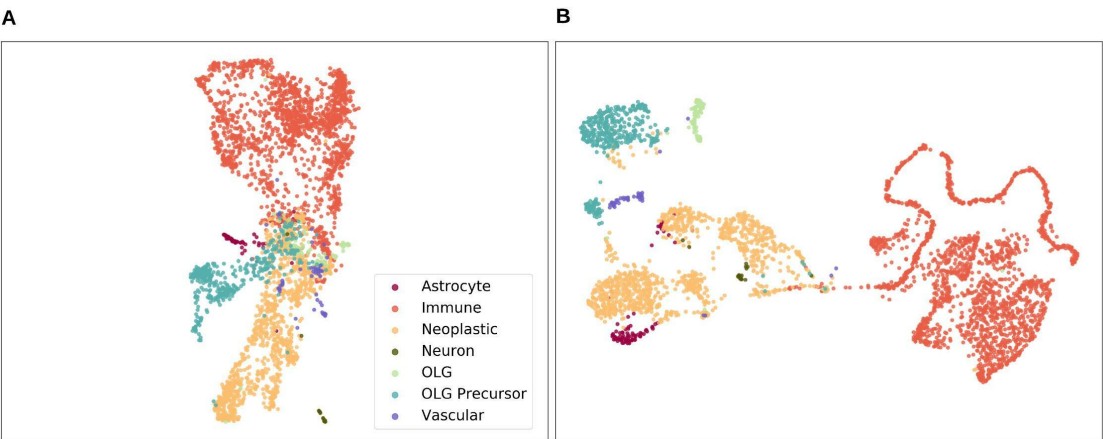

Figure 3: Further analysis of top features selected by AdaSTAMPS. (A) Visualization of 2-dimensional embeddings obtained from UMAP that is trained using all $M = 28805$ features. (B) Visualization of 2-dimensional embeddings obtained from UMAP that is trained using the top seven features selected by AdaSTAMPS (above default threshold $\pi_{\mathrm{thr}} = 0.5$). Note that every point (cell) is colored according to its class label.

## 4    Discussion

We have developed general meta-algorithms for feature selection named STAMPS and AdaSTAMPS that leverage random minipatches as well as adaptive feature sampling schemes. We empirically demonstrate that our approaches, especially AdaSTAMPS, have superior feature selection performance and computational efficiency for a variety of synthetic and real-world scenarios. The main advantage of our methods is that these are practical solutions that we can employ with a wide variety of existing learning algorithms and feature selection strategies to identify statistically accurate features in a computationally efficient manner, even in high-dimensional and high-correlation settings, thus enhancing interpretability. Our proposed feature selection frameworks are implemented in our `Python` package `minipatch-learning`, available at https://github.com/DataSlingers/minipatch-learning.

While we develop two effective adaptive feature sampling schemes in this work, exploring other types of adaptive learning techniques to be used in conjunction with AdaSTAMPS is an interesting future direction. In addition, we would like to develop theoretical frameworks for analyzing and understanding the theoretical underpinning of the more sophisticated AdaSTAMPS method in future work.

Overall, we have proposed practical and computationally efficient feature selection methods with strong empirical performance and developed a user-friendly, open-source software package in this work.

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

## A  Probabilistic Adaptive Feature Sampling Scheme

In Algorithm 3, we give our probabilistic adaptive feature sampling scheme. Here, $k$ is the iteration counter, $E$ denotes the total number of burn-in epochs. We use the shorthand AdaSTAMPS (Prob) to denote the specific instance of AdaSTAMPS that employs Algorithm 3 as its adaptive feature sampling scheme.

Similar to our exploitation and exploration scheme (Algorithm 2 in the main paper), AdaSTAMPS (Prob) first explores the entire input feature space during the burn-in stage. After $E$ burn-in epochs, Algorithm 3 transitions to the adaptive stage during which it adaptively updates the sampling probabilities for all features. Specifically, features with higher selection frequencies, which are likely to be the true signal features, have relatively higher probabilities of being sampled into subsequent minipatches. However, since this is a probabilistic sampling scheme, Algorithm 3 still adaptively explores the remaining feature space because many features with relatively lower selection frequencies still have non-zero probabilities of being sampled into minipatches.

---

**Algorithm 3** Adaptive Feature Sampling Scheme: Probabilistic (Prob)

---

**Input:** $k$, $M$, $m$, $E$, $\{\hat{\Pi}_j^{(k-1)}\}_{j=1}^M$

**Initialize:** $G = \lceil \frac{M}{m} \rceil$, $\mathcal{J} = \{1, \ldots, M\}$

**if** $k \leq E \cdot G$ **then** // Burn-in stage

    **if** $\mathrm{mod}_G(k) = 1$ **then** // A new epoch

        1) Randomly reshuffle feature index set $\mathcal{J}$ and partition into disjoint sets $\{\mathcal{J}_g\}_{g=0}^{G-1}$

    **end if**

    1) Set $F_k = \mathcal{J}_{\mathrm{mod}_G(k)}$

**else** // Adaptive stage

    1) Update feature sampling probabilities using information from previous iterations:

$$\hat{p}_j^{(k)} = \frac{\hat{\Pi}_j^{(k-1)}}{\sum_{j=1}^M \hat{\Pi}_j^{(k-1)}} \quad \text{for} \quad j \in \{1, \ldots, M\}$$

    2) Sample $m$ features $F_k \subset \{1, \ldots, M\}$ without replacement according to the updated feature sampling probabilities $\{\hat{p}_j^{(k)}\}_{j=1}^M$

**end if**

**Output:** $F_k$

---

# B   More on Practical Considerations

## B.1   Data-Driven Rule to Choose $\pi_{\mathrm{thr}}$

While there are many possible data-driven approaches to choose the user-specific threshold $\pi_{\mathrm{thr}} \in (0, 1)$, we propose a very simple one in Algorithm 4 that we found to be effective in practice. Here, $\{\hat{\Pi}_j^{(K)}\}_{j=1}^M$ is the set of feature selection frequencies from the last iteration of STAMPS or AdaSTAMPS, and $\hat{\mathrm{SD}}$ denotes the sample standard deviation. Intuitively, Algorithm 4 tries to find all gaps (local minima of density) among $\{\hat{\Pi}_j^{(K)}\}_{j=1}^M$ in order to choose a threshold that separates features with relatively high selection frequencies from those with low selection frequencies. Algorithm 4 does so by first fitting a Gaussian kernel density estimator (KDE) to the selection frequencies and then setting $\pi_{\mathrm{thr}}$ to the smallest local minima, if any.

---

**Algorithm 4** KDE-Based Data-Driven Rule to Choose $\pi_{\mathrm{thr}}$

---

**Input:** $\{\hat{\Pi}_j^{(K)}\}_{j=1}^M$

1) Fit a Gaussian KDE to $\{\hat{\Pi}_j^{(K)}\}_{j=1}^M$:

$$\hat{f}_h(x) = \frac{1}{M} \sum_{j=1}^M \exp\Big(\frac{(x - \hat{\Pi}_j^{(K)})^2}{2h^2}\Big), \forall x \in [0,1]$$

where the bandwidth of the Gaussian kernel $h = \hat{\mathrm{SD}}(\{\hat{\Pi}_1^{(K)}, \hat{\Pi}_2^{(K)}, \ldots, \hat{\Pi}_M^{(K)}\})$

2) Use method of inflection to find all local minima:

$$\mathcal{D} = \{x \in [0,1] \text{ s.t. } \mathrm{sign}(\hat{f}_h(x) - \hat{f}_h(x - \epsilon)) = -1, \mathrm{sign}(\hat{f}_h(x + \epsilon) - \hat{f}_h(x)) = 1\}$$

for any $\epsilon > 0$
**if** $|\mathcal{D}| \geq 1$ **then** // Find at least one local minima

    Set $\pi_{\mathrm{thr}} = \min(\mathcal{D})$

**else** // No local minima is found

    Set $\pi_{\mathrm{thr}} = 0.5$
**end if**
**Output:** $\pi_{\mathrm{thr}}$

---

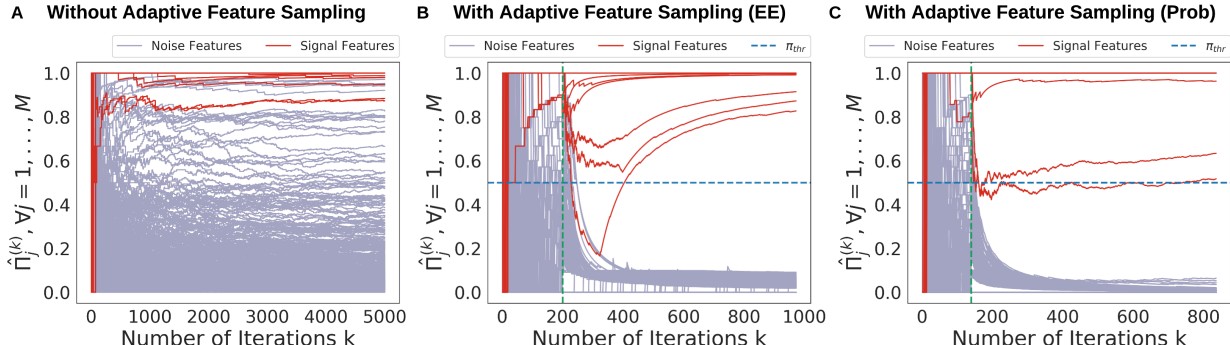

Figure 4: Performance of our approaches on a challenging feature selection task in the regression setting with highly correlated features (as described in Section B.3). (A) Feature selection frequencies by STAMPS versus number of iterations; (B) Feature selection frequencies by AdaSTAMPS (EE) versus number of iterations with the green dotted line denoting the end of the burn-in stage; (C) Feature selection frequencies by AdaSTAMPS (Prob) versus number of iterations. This suggests that well-designed adaptive feature sampling schemes help AdaSTAMPS correctly select all true signal features (red) while discarding all noise features (gray) within much fewer iterations, thus potentially further boosting both feature selection accuracy and computational efficiency in practice.

## B.2  Empirical Illustrations of Advantages of Adaptive Feature Sampling

We also empirically illustrate that using well-designed adaptive feature sampling not only increases computational efficiency but also seems to further improve feature selection accuracy in practice. In Figure 4, we display selection frequencies of all features versus the number of iterations for STAMPS, AdaSTAMPS (EE), and AdaSTAMPS (Prob) on a challenging feature selection task. We see that both AdaSTAMPS (EE) and AdaSTAMPS (Prob) correctly identify all true signal features while discarding all noise features within 1000

iterations. On the other hand, STAMPS without adaptive feature sampling converges slower and is unable to clearly separate a handful of true signal features from the noise features. Details of this empirical study are given in Section B.3.

### B.3 Additional Empirical Studies to Investigate Effects of Minipatch Size

Here, we empirically investigate how feature selection accuracy and runtime of our meta-algorithm vary for various minipatch sizes (i.e. $n$ and $m$ values). We take a subset of the TCGA data that contains 450K methylation status of the top $M = 2000$ most variable CpGs for $N = 2000$ patients as the data matrix $\mathbf{X} \in \mathbb{R}^{N \times M}$. Following similar procedures to Section 3.1 of the main paper, we create the true feature support $S$ by randomly sampling $S \subset \{1, \ldots, M\}$ with $|S| = 10$. The $M$-dimensional sparse coefficient vector $\boldsymbol{\beta}$ is generated by setting $\beta_j = 0$ for all $j \notin S$ and $\beta_j$ is chosen independently and uniformly in $[-3/b, -2/b] \cup [2/b, 3/b]$ for all $j \in S$. Finally, the response vector $\mathbf{y} \in \mathbb{R}^N$ can be generated by $\mathbf{y} = \mathbf{X}\boldsymbol{\beta} + \boldsymbol{\epsilon}$ where the noise vector $(\epsilon_1, \ldots, \epsilon_N)$ is IID $\mathcal{N}(0, 1)$. The positive real number $b$ is chosen such that the SNR is fixed at 5. This synthetic data set is also used for the experiment in Figure 4 in the previous section.

To study how performance varies with minipatch size ($n$ and $m$), we run AdaSTAMPS (EE) using thresholded OLS as the inner base selector for various $n$ and $m$ values. Specifically, we take a sequence of $m$ values that are integer multiples of $|S|$ and then pick $n$ relative to $m$ such that $m/n \in \{0.1, 0.2, \ldots, 0.8\}$. Feature selection accuracy in terms of F1 Scores and computational time are reported for various $n$ and $m$ values in Figure 5. We see that our method is fairly robust to the choice of minipatch size for it has largely stable feature selection performance for a sensible range of $n$ and $m$. Taking a closer look, we can see that there is a slight trade-off between feature selection accuracy and computational efficiency - using larger minipatches (i.e. larger $m$) generally results in better final selection accuracy but also leads to longer overall runtime. In general, we found taking $m$ to well exceed the number of true signal features (e.g. 3-10 times the number of true signal features $|S|$) and then picking $n$ relative to $m$ so that it well exceeds the sample complexity of the base feature selector used in the meta-algorithm can strike a good balance between accuracy and computational efficiency in practice.

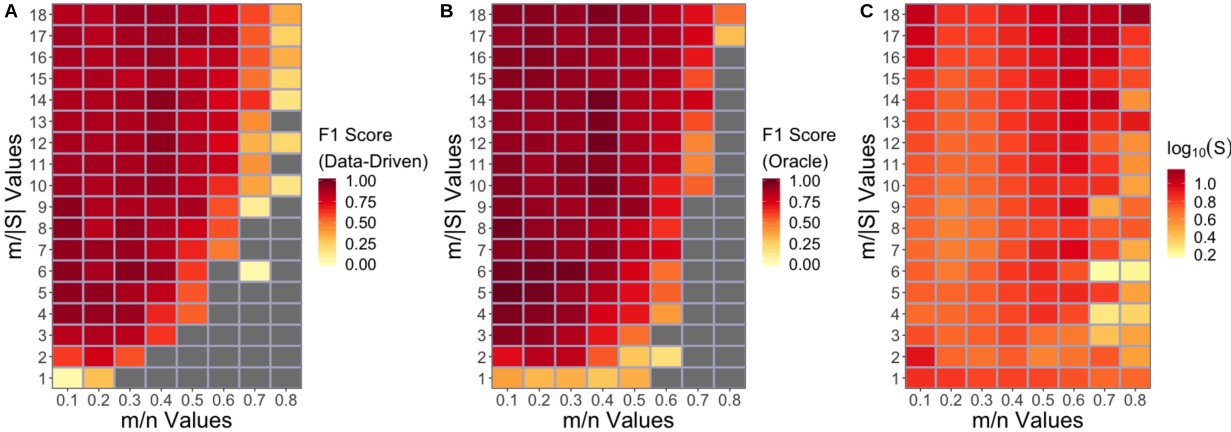

Figure 5: Effects of minipatch size. We show how feature selection accuracy and computational time of AdaSTAMPS (EE) vary with minipatch sizes in terms of $m/|S|$ and $m/n$, where $|S|$ is the cardinality of the true feature support. F1 Scores and computational time are averaged over 5 replicates. (A) Feature selection accuracy in terms of F1 Score with data-driven $\pi_{\mathrm{thr}}$; (B) Feature selection accuracy in terms of F1 Score with oracle model selection; and (C) Computational time on $\log_{10}(\text{second})$ scale. Note that gray cells indicate NA's that result from poor selection performance. We see that our method has largely stable performance for a sensible range of $n$ and $m$.

## C   Additional Materials for Empirical Studies

### C.1   Data-Driven and Oracle Model Selection Procedures for Various Methods in Empirical Studies

In this section, we discuss details of data-driven and oracle model selection procedures that we use for our empirical studies in Section 3.1 of the main paper.

For the Lasso, data-driven model selection approaches such as the 10-fold cross-validation with 1 standard error (SE) rule (Hastie et al., 2009) and the extended BIC (eBIC) rule (Chen & Chen, 2008) are employed to determine the optimal regularization parameter, resulting in $\lambda_{1SE}$ and $\lambda_{eBIC}$, respectively. For univariate filter, the data-driven rule keeps all features that are significantly associated with the response with FDR controlled at the default level of 0.05. Additionally, 10-fold cross-validation is carried out for RFE to choose the best tuning hyperparameters and the Bonferroni procedure (Giurcanu, 2016) is used with thresholded OLS to determine the selected features in a data-driven manner. The user-specific threshold $\pi_{thr}$ is set to 0.5 for STAMPS, AdaSTAMPS (EE), AdaSTAMPS (Prob), CPSS, and Randomized Lasso. Following the default settings in `scikit-learn` (Pedregosa et al., 2011), we use a geometric sequence $\Lambda \in (0.001\lambda_{max}, \lambda_{max})$ as the set of candidate regularization parameters for all Lasso-based methods, where $\lambda_{max}$ is the minimum amount of regularization such that all estimated coefficients become zero. $|\Lambda|$ is set to the default 100 for simulation scenarios S1-S2 and to 10 for scenario S3. For AdaSTAMPS (EE), we also use default settings with $E = 10$ burn-in epochs and $\pi_{active} = 0.1$. For AdaSTAMPS (Prob), $E$ is also set to 10. Additionally, we use minipatch size $(n = 180, m = 60)$ for Scenario S1-S3 in Section 3.1. Lastly, for STAMPS, AdaSTAMPS (EE), AdaSTAMPS (Prob), the stopping criterion parameters $\tau_u$ and $\tau_l$ are set to 60 and 30, respectively.

Assuming knowledge of the number of true signal features $|S|$, oracle model selection is also applied to every method. Specifically, oracle model selection is applied to STAMPS, AdaSTAMPS (EE), AdaSTAMPS (Prob), CPSS, and Randomized Lasso by keeping the top $|S|$ features with the highest selection frequencies. For the Lasso, oracle selection keeps the top $|S|$ features with the largest magnitude in terms of coefficient estimates at $\lambda_{1SE}$ and $\lambda_{eBIC}$, respectively. For univariate filter, the top $|S|$ features with the smallest p-values are kept and the top $|S|$ features with the largest magnitude in terms of OLS coefficient estimates are selected for thresholded OLS. Furthermore, RFE is carried out until exactly $|S|$ features are left for oracle model selection.

### C.2   Additional Details about the Real-World Classification Data

In this section, we describe the training procedures used in Section 3.2 of the main paper. For the experiments in Figure 2A of the main paper, our AdaSTAMPS (EE) fits decision tree classifiers as base selectors to many random minipatches and keeps 10 features with the highest Gini importance from every minipatch. Features are then ranked based on their selection frequencies from AdaSTAMPS (EE), which are computed by taking an ensemble of selection results over these minipatches. We note that we use default settings for AdaSTAMPS (EE) with $E = 10$ burn-in epochs and $\pi_{active} = 0.1$, and a fixed minipatch size $(n = 600, m = 200)$ for experiments in Section 3.2. For comparisons, we also fit a RF classifier to the full training data and obtain two sets of ranks for all features based on the Gini impurity-based importance and permutation-based importance, respectively. Moreover, we obtain another set of ranks for features using a RFE with RF as base estimator on the full training data.

For the experiments in Figure 2B of the main paper, our AdaSTAMPS (EE) fits unregularized multinomial regressors as base selectors to many random minipatches and keeps 10 features with the largest coefficient magnitudes from every minipatch. Features are then ranked based on their selection frequencies from AdaSTAMPS (EE), which are computed by taking an ensemble of selection results over these minipatches. For comparison, we fit a elastic-net regularized multinomial regressor to the full training data (with cross-validation to choose optimal tuning parameter values) and obtain ranks for all features based on corresponding coefficient magnitudes. Last but not least, univariate filtering is applied to the training data, producing another set of ranks for features based on resulting p-values.

