# OpenReview forum: "Feature Selection for Huge Data via Minipatch Learning"
_TMLR — Withdrawn by Authors_

### Review · Reviewer_DY3s · 2023-03-02

**Summary Of Contributions:**

This paper proposes a meta-algorithm for performing feature selection with datasets that contain many rows and many columns. Their approach is a wrapper around any existing feature selection algorithm (the "base selector"), and it works by repeatedly running the base selector on subsampled versions of the dataset (minipatches), aggregating the results, and returning the set of features that are selected most consistently.

The authors consider a number of practical concerns about the algorithm, including several hyperparameters, how to choose the number of sampled subsets, and whether it is possible to accelerate the algorithm's convergence. They propose reasonable heuristics for some of these choices, and also propose an adaptive procedure to accelerate convergence by focusing on more important features.

The experiments show that this procedure is more effective than several baselines (e.g., lasso, thresholded OLS): the authors observe more accurate recovery of the truly important features with synthetic linear regression datasets, and more accurate prediction accuracy with the top features using a real dataset.

**Audience:**

No

**Broader Impact Concerns:**

I don't see any ethical concerns with this work.

**Claims And Evidence:**

No

**Requested Changes:**

I would request that the authors expand their experiments section. This would help provide more convincing evidence that STAMPS works robustly across datasets, hyperparameters values, and base selectors. For example:
- Include additional baselines
- Investigate the choice of base selector, and its hyperparameters
- Include more real datasets

I would also request that the authors clarify several important points:
- What is the difference between STAMPS and stability selection? Would it be incorrect to conclude that the only new idea here is the adaptive procedure in Algorithm 2?
- Discuss why you think it is that Algorithm 2, in addition to reducing the running time, improves performance. It's not clear from the text why we would expect this to be the case, so the results are somewhat confusing.
- Discuss more thoroughly why some of the hyperparameters are safe to set to the suggested default values. I can see, for example, that $\pi_{thresh}$ may be safe to set to 0.1 because it's relatively conservative in ignoring irrelevant features, but this isn't actually explained.
- Discuss whether any error guarantees similar to those from stability selection can be provided here.

**Strengths And Weaknesses:**

Strengths:
- The algorithm is simple and speeds up computation.
- It seems to work quite well in the experiments shown here.
- The adaptive sampling idea not only speeds up computation, but yields better results (only illustrated via feature selection accuracy in Figure 1).

Weaknesses:
- There's minimal methodological innovation, Algorithm 1 seems almost identical to stability selection as described by Meinshausen & Bühlmann (2010). If I'm not mistaken, stability selection considers random sampling over both rows and columns ("minipatches"), so the only new idea here may be the adaptive approach in Algorithm 2.
- There's no theoretical analysis of why this approach should work, the only evidence is empirical. I'm not sure any formal analysis is possible here, particularly given all the moving parts in the adaptive algorithm, so I don't hold it against the authors that they didn't try. But the lack of analysis makes the contribution somewhat shallow (note that stability selection is accompanied by some guarantees, e.g., regarding the number of false selections).
- In my view, the authors downplay the number of hyperparameters. In addition to the number of rows $n$, the number of columns $m$ and the threshold $\pi_{thresh}$ used to make selections, there's also the convergence criterion (how to determine $K$), the schedule for $\gamma$, $E$ for the burn-in duration, and a threshold $\pi_{active}$ for the adaptive algorithm. I get that some of these have heuristics that work well in these experiments, but there isn't a clear argument that those settings will work well across arbitrary datasets. And all this ignores the crucial choice of base selector, as well as any additional hyperparameters associated with that method. This is arguably an overwhelming number of hyperparameters, and the experiments don't look closely at most of them.
- The experiments aren't very thorough. As mentioned above, the authors don't look into several hyperparameters. Perhaps most importantly, they only test one base selector. How sensitive is the method to the choice of base selector? How about to the hyperparameters for the base selector? And would it be possible to include >1 real datasets?
- Some of the model selection procedures seem odd. In the "oracle" setting where you assume knowledge of the true number of important features $|S|$, why not simply fit Lasso to get the desired number of features (e.g., using LARS)? And for the univariate filter method, perhaps this is what you're doing already, but it seems like the simplest approach would be to retain the top $|S|$ features with the highest correlation.
- The baseline methods all look very old, are there no more recent methods than this? I'm not an expert on high-dimensional feature selection so I'm not sure, but at the very least some other classical methods would be greedy forward selection (aka orthogonal matching pursuit), matching pursuit, and perhaps elastic net. For the Figure 2 results with the random forest, how about mean absolute TreeSHAP values? And it's a different type of method based on deep learning, but the Concrete Autoencoder has shown some success in high-dimensional selection.

Presentation issues:
- Perhaps there's a better way to phrase this: "To avoid a forever STAMPS situation"
- The emphasis on the software package was a bit odd to me. The bottleneck in this algorithm should be the repeated calls to the base selector, right? The logic on top of that seems very simple. In Algorithm 1 for example, all you need to do is count how many times a feature was randomly sampled and how many times it was selected.

---

### Review · Reviewer_yWDN · 2023-03-07

**Summary Of Contributions:**

The authors propose a meta algorithm for feature selection based on estimating the support for important features of a base feature selector on a randomly sampled minipatch, i.e. a much smaller random sample of the columns and of the rows of the data matrix.

**Audience:**

Yes

**Broader Impact Concerns:**

No concerns.

**Claims And Evidence:**

No

**Requested Changes:**

Using only the information available in very small patches is going to end up in having to pay a price in bias somewhere, but the author do not analyse where the boundaries of the approach are. For example, when features are informative only when considered jointly, the chance for all of them of being selected in the same mini-patch decrease exponentially. Because the authors analyse only an artificial  regression task where each feature contributes 'additively' to the target, these issues are ignored. The authors should investigate the approach limits and offer empirical evidence of the robustness of the approach in these limit cases [critical].

The authors propose a robust meta algorithm that is applicable to any base feature selection algorithm, however there clearly can exist important effects between the hyper-parameters of the proposed STAMP (i.e. n, m, \pi_thr) and the hyper-parameters of the base algorithms, which would affect the relative robustness of the approach w.r.t. hyper-parameters selection. In practice the authors should at least show how the performance is affected when changing the underlying base selector using a few common feature selection algorithms [critical].

The authors claim in Section 3 that the approach is not limited to linear or regression settings, but do not offer support for the claims: it would be useful to see how the approach performs in classification problems and in non-linear regression settings.
We note that evaluating the fraction of features identified w.r.t the features that were actually used in building the (artificial) target does not take into account the presence of correlated features in the data that if selected would yield a similar target signal. The authors should discuss this issue and propose a fearer evaluation approach [non-critical].

The approach is clearly efficient, however the computational complexity of the approach is evaluated only w.r.t. the signal to noise ratio, while another (equally interesting) dimension of variability would be the actual number of selected features to search for, i.e. the |S| size [critical].

As for the real world case, please note that cell types are commonly defined by biologists according to the presence of marker genes being highly expressed. To make a more challenging task consider the binary classification problem of disease vs healthy cell state where the target is not based on human-centric notions [non-critical].



**Strengths And Weaknesses:**

Pos:
- the computational complexity is significantly reduced while at the same time not impacting on the quality of the selection accuracy.
- well documented implementation available

Neg:
- some more evidence has to be shown to demonstrate the generality and effectivity of the approach.
- the limitations of the approach are not analysed in depth.

---

### Review · Reviewer_Cj4h · 2023-03-17

**Summary Of Contributions:**

The paper proposes three algorithms for feature selection in very large dataset that are based on the idea of running a base feature selection algorithm on subsets of both features and samples (so-called mini-patches). The features that are selected more than $\pi_{thr}$% of the time against the other feature in the mini-patches are returned. The first algorithm, STAMP, simply samples mini-patches independently of each other, while the other two algorithms, adaSTAMP(EE) and (Prob), sample them adaptively, by taking into account the features that were selected in the previous rounds. The main motivation for the approaches is to improve computing times in the case of very large datasets. Experiments are conducted on three artificial linear regression problems and on a real dataset in the biomedical domain. STAMP and adaSTAMP are shown to perform very well, both in terms of accuracy of the retrieved features and computing times, against several competitors.


**Audience:**

Yes

**Broader Impact Concerns:**

I have no concerns about the ethical implications of the work.

**Claims And Evidence:**

No

**Requested Changes:**

I definitely think the ideas in the paper are worth explaining. But the paper needs more work. All my comments above should be taken into account.
- The work should be more clearly positioned with respect to related works.
- The proposed algorithms should be better and more formally motivated and their limitation better highlighted.
- Experiments should be extended to include more diverse scenarios (to study the impact of the number of relevant features, of the base feature selection method, of correlation patterns) and to include more systematic experiments on a larger range of problems.


**Strengths And Weaknesses:**

The idea of training feature selection algorithms on mini-patches is appealing. It mimics similar algorithms that have been proposed to improve predictive performance or computing times of base learners in the ensemble learning literature. The idea is particularly appealing from a computational point of view, as feature selection methods have a high computational complexity with respect to the number of features. It has also the potential to improve stability of the selected features by the ensembling effect. The two proposed adaptive algorithms are also valid ideas that deserve to be studied.

The specific approaches investigated in the paper are sufficiently novel, despite being close to several works. The discussion of related works in the paper is however not totally satisfactory in my opinion.

1) The authors mention in Section 2.2 that their approach is inspired by stability selection techniques for linear model. These works are however merely cited without being explicitly linked to (ada)STAMPS. How do STAMPS and adaSTAMPS differ from these works? For example, (Staerk et al., 2021) seems to also propose an adaptive sampling scheme. How different is it from adaSTAMPS' two schemes?

2) The related works in Section 1.1 about feature selection are mostly focused on linear methods. There are other papers that explore similar ideas with other base feature selection methods. For example:
[1] Saeys et al., Robust Feature Selection Using Ensemble Feature Selection Techniques. ECML/PKDD 2008.
[2] Draminski et al. Monte carlo feature selection for supervised classification. Bioinformatics, 24(1):110–117, 2008.
[3] Sutera et al., Random subspace with trees for feature selection under memory constraints, AISTATS 2018.
[4] Tian and Feng, RaSE: Random Subspace Ensemble Classification. JMLR 2021.
The first two works propose ideas similar to Algorithm 1 in the paper, while [3] and [4] seem to propose adaptive sampling schemes. The algorithm in [3] is very close to Algorithm 2, although specific to random forests.

3) The paper also do not talk at all about feature selection methods for very high-dimensional input spaces. This literature should be looked at. Just to give an example: Aghazadeh et al.  MISSION: Ultra Large-Scale Feature Selection using Count-Sketches. ICML 2018.

While I think the ideas are worth exploring, my main problem with the algorithms is their very heuristic nature. Here are some problems/thoughts related to that:

4) The feature selection problem that the paper wants to address is not well formulated in the first paragraph of the introduction. For a given problem, there are many $S$ such that the output is independent of $S^c$ knowing $S$. Are the authors interested in the smallest such $S$? Are they looking to put in $S^c$ only features that are fully irrelevant? Without a more formal problem definition, it's difficult to assess whether the approaches are successful.

5) Correlation is discussed a lot in the paper and The good performance of (ada)STAMPS is explained in the paper notably by the fact that the method "breaks up strong correlations among features". What does it mean and why is it a good thing? In general the effect of feature correlation on the algorithms should be better analysed, theoretically and empirically. Let's assume that there are two features that are strongly correlated and only one of them appears in the true model. To detect such correlation pattern, the two features should be available to the feature selection algorithm. But if $m$ is small with respect to the total number of features, the chance that these two features will be picked together might be small and both of them will be considered as good features, including the irrelevant one.

6) The algorithm depends on many hyper-parameters of the algorithms (patch sizes, m and n, number of iterations of the algorithm, probability threshold, the base feature selection algorithm that has its own hyper-parameters, etc.) and clearly they should affect which and how many features will be found by the algorithm. In real problems, there is no possibility to use cross-validation to tune these hyper-parameters. Without a deeper understanding of what the algorithms are actually doing or are expected to produce (see my first point), it's difficult to understand how to set these hyper-parameters and to interpret the retrieved features. For example, the authors say in Appendix B.3 that m should be selected to well exceed the number of true signal features. Why is it necessary to get good performance?  And how do we know the number of true signal features for a real problem?

7) The question of the choice of the base feature selector is overlooked in the paper. Many statements about the method, e.g., its ability to deal with correlated features and its efficiency, are made without any reference to this base feature selector. I believe its properties should affect strongly the selected features. The authors should discuss which feature selection methods are expected to benefit the most from their wrappers and which properties the feature selection methods should satisfy for STAMPS or adaSTAMPS to produce good results.

8) My understanding of the two adaptive approaches is even more limited. They are motivated as ways to fasten the non-adaptive approach and to make it more accurate. Why would they be necessarily more accurate?  I don't find the way the adaptive algorithms are motivated at the end of Section 2.3 to be convincing.

This lack of theory or principled motivation could partially be compensated by the good empirical performances. And indeed, the empirical results are quite good and show that the algorithms have strong potentials. I like in particular the experiments in Section B.3 and the plots in Figure 4. The experiments have however important limitations:

9) In all artificial problems considered, the ground truth is linear and the number of relevant features is fixed to 20, which is very restrictive. How is the performance affected by non-linearity, by the number of relevant features, by the correlation pattern, etc. ?

10) A single real-world problem is tackled. There are several feature selection benchmarks that could be used to more systematically assess the methods (e.g. scikit-features).

11) Hyper-parameter tuning of (ada)STAMPS is very unclear. While all other methods have their parameters tuned by cross-validation (because they learn a regression model), hyper-parameters of STAMP seems to have been set fully manually on a per problem basis. The study in B.3 is interesting but it is done on a single problem. Does this analysis generalize to other problems (non-linear, with more relevant features, with other base feature selection algorithms, etc.)?

12) Related to point (7), I think that (ada)STAMPS should have been tested with different base feature selection techniques. What is the best combination? Are there feature selection method that benefit more from mini-patches, in terms of accuracy or computing times? Does (ada)STAMP improve all base feature selection methods like it does for OLST?

The paper is well written and structured. A few minor problems:
- I find the general tone of the paper to be too positive about the proposed methods. For example, in Section 1.1, the authors systematically mention the drawbacks of existing methods, but never actually show that they are addressed by their methods. There is also no discussion at all of the limitations of their work.
- Algorithm 2 describes one iteration of the global algorithm, while Algorithm 1 gives the full procedure. Why not describe the full procedure in Algorithm 2?
- Notations [M] is used in Algorithm 1 but not in Algorithm 2.
- "Last but no least" sounds weird in a scientific paper.

Other minor comments:
- I'm not sure we can talk about "huge data" for the dataset sizes considered in the paper, especially in terms of the number of samples.
- Section 2.1 is not very useful. Why not include this into the related work section.
- To get a better assessment of the feature ranking provided by the different feature selection methods, I would suggest to use an AUC or AUPR score instead of F1-score (as a complement to the oracle score).
- In 3.2: using random forests as the downstream classifier to assess the selected features is typically not a good idea. Indeed, this classifier is rather robust to the introduction of irrelevant features, which might not penalize enough feature selection methods that makes false positive predictions.

---

### Note · Authors · 2023-03-30

I have read and agree with the venue's withdrawal policy on behalf of myself and my co-authors.